# Comparison of Foraging Interactive D-prime and Angular Indication Measurement Stereo with different methods to assess stereopsis

**Sonisha Neupane** *, Jan Skerswetat, Peter J. Bex

Department of Psychology, College of Science, Northeastern University, Boston, Massachusetts, United States of America

* s.neupane@northeastern.edu

**Data Availability Statement:** All data files are available from the Zenodo database (accession number 10.5281/zenodo.1068886).

## Abstract

### Purpose

Stereopsis is a critical visual function, however clinical stereotests are time-consuming, coarse in resolution, suffer memorization artifacts, poor repeatability, and low agreement with other tests. Foraging Interactive D-prime (FInD) Stereo and Angular Indication Measurement (AIM) Stereo were designed to address these problems. Here, their performance was compared with 2-Alternative-Forced-Choice (2-AFC) paradigms (FInD Stereo only) and clinical tests (Titmus and Randot) in 40 normally-sighted and 5 binocularly impaired participants (FInD Stereo only).

### Methods

During FInD tasks, participants indicated which cells in three 4*4 charts of bandpass-filtered targets (1,2,4,8c/˚ conditions) contained depth, compared with 2-AFC and clinical tests. During the AIM task, participants reported the orientation of depth-defined bars in three 4*4 charts. Stereoscopic disparity was adaptively changed after each chart. Inter-test agreement, repeatability and duration were compared.

### Results

Test duration was significantly longer for 2-AFC (mean = 317s;79s per condition) than FInD (216s,18s per chart), AIM (179s, 60s per chart), Titmus (66s) or RanDot (97s). Estimates of stereoacuity differed across tests and were higher by a factor of 1.1 for AIM and 1.3 for FInD. No effect of stimulus spatial frequency was found. Agreement among tests was generally low ($R^2$ = 0.001 to 0.24) and was highest between FInD and 2-AFC ($R^2$ = 0.24; $p<0.01$). Stereoacuity deficits were detected by all tests in binocularly impaired participants.

### Conclusions

Agreement among all tests was low. FInD and AIM inter-test agreement was comparable with other methods. FInD Stereo detected stereo deficits and may only require one condition

**Funding:** This project was supported by National Institutes of Health (www.nih.gov) (grant R01 EY02971 to PJB). The funders had no role in study design, data collection and analysis, decision to publish, or preparation of the manuscript.

**Competing interests:** I have read the journal's policy and the authors of this manuscript have the following competing interests: FInD and AIM technologies are disclosed as provisional patented (AIM) and pending patent (FInD) and held by Northeastern University, Boston, USA. FInD title: Method for visual function assessment; Application PCT/US2021/049250 AIM title: Self-administered adaptive vision screening test using angular indication; Application PCT/US2023/012959 JS and PJB are founders and shareholders of PerZeption Inc, which has an exclusive license agreement for FInD and AIM with Northeastern University. SN declares that she has no conflict of interest. This does not alter our adherence to PLOS ONE policies on sharing data and materials.

to identify these deficits. AIM and FInD are response-adaptive, self-administrable methods that can estimate stereoacuity reliably within one minute.

## Introduction

Stereopsis is a critical function of the human visual system and is a cornerstone of perception across many species. Impairment of stereopsis often indicates the presence of a visual disorder during development or neurodegenerative disease [1, 2]. Assessing and monitoring stereopsis is therefore critical in detecting and managing a range of disorders, however, booklet-based clinical methods for measuring stereoacuity have several problems including: low sensitivity to changes of stereoacuity; coarse resolution [3]; poor agreement across tests [4, 5]; poor test-retest repeatability, especially when stereoacuity is low [6] memorization artefacts; monocular cues [7, 8]; assumptions based on testing distance and interpupillary distance and less flexibility on testing distance.

Poor agreement across tests [4, 5, 9] could arise from differences in the task, stimulus properties and display technology. Stereoacuity thresholds can also depend on the direction of the disparity (crossed/near or uncrossed/far depth), but most tests only measure one direction (typically crossed disparity). Different tests use different stimuli, but stereoacuity depends on spatial structure [10, 11], eccentricity [12] and may be differentially affected by degraded image quality (e.g., from refractive error, cataract, or amblyopia) [13].

Many adaptive computer-based methods, such as Alternative Forced Choice (AFC) tasks, address the above problems of inaccuracy and imprecision. Although adaptive procedures can generate more sensitive measures of threshold performance, these tests are time-consuming [14, 15] and the repeated administration of below threshold stimuli can be frustrating for naïve participants.

To address these problems, we adapted two novel computer-based methodologies for the assessment of stereoacuity: Foraging Interactive D-prime (FInD) [16] and Angular Indication Measurement (AIM) [17]. Both methods are 1) computer-based, thus they are deployable on a range of devices; 2) randomized to avoid memorization artifacts across repeated tests [18]. 3) stimulus-agnostic, thus can be used to display local (random dot-defined) or global (ring) targets, with broad-band or narrow-band stimuli, and with either crossed, uncrossed, or mixed disparity; 4) self-administered, removing the need for a clinical examiner and enabling home testing; 5) response-adaptive, allowing accurate measurement of stereoacuity in people with low or high stereopsis and precise measurement of small differences in stereoacuity; 6) simple with user-friendly tasks and interfaces that can be completed by participants with a range of ages or cognitive functional levels [19]; 7) generalizable, using the same fundamental task for the assessment of multiple visual functions (e.g. acuity, color, contrast, motion and form sensitivity, among others), minimizing the number of tasks the participant is required to learn.

The two studies reported here were performed as proof-of-concept studies for 2 novel methods to measure stereoacuity: In Study One, we introduce FInD Stereo and its features. We compare estimates of stereoacuity, test duration, inter-test reliability, and repeatability of FInD with standard 2-AFC methods, and clinically used tests (Randot and Titmus) in stereotypical and atypical participants. We also use the FInD method to compare different spatial properties of stereo-inducing stimuli and examine their effect on stereoacuity. In Study Two, we introduce AIM Stereo and its features, then we compare AIM Stereo against the above-mentioned clinical tests using the same outcome measures.

## Study 1 FInD Stereo—Methods

The study was approved by the Institutional Review Board at Northeastern University (14-09-16) and followed the guidelines of the Declaration of Helsinki. The recruitment period was from November 29, 2021 to October 30, 2022. Informed written consent was obtained from all participants prior to the start of the experiment. The participants were the staff of the research laboratory and undergraduate students who completed the study for course credit.

### Participants

20 normally sighted and 5 binocularly impaired (3 with self-reported amblyopia, 1 with strabismus and 1 with strabismus and amblyopia) adults participated in Study 1. One participant was excluded due to poor vision. Participants' details for Study 1 are provided in Table 1.

### Stimuli and procedure

Stimuli were generated using Mathworks MATLAB software (Version 2021b), the Psychtoolbox [20–22] and were presented on a gamma-corrected 32" 4K LG monitor with maximum luminance of $250/m^2$ and screen resolution of 3840 x 2160 and 60Hz at 80 cm viewing distance. A chinrest was used to maintain the viewing distance. Red-blue anaglyph glasses were used to present the stimuli to each eye dichoptically. Horizontal disparity was used to induce stereo-depth.

In Study 1, participants performed six different tests: two FInD stereoacuity tasks, two 2-AFC stereoacuity tasks, and two clinical tests (Randot and Titmus) in randomized order. The time taken for participants to complete the self-administered FInD and 2-AFC tests was recorded by the test computer, the time taken to complete the examiner-administered clinical tests was recorded with a stopwatch.

### FInD Stereo

FInD [16] is a self-administered paradigm in which stimuli are displayed over one or more charts, each containing a grid of N cells (here N = 16), a random subset of which (a uniform

**Table 1. Demographic and optometric summary of the cohorts for Study 1.**

|  | Binocularly normal | Binocularly impaired |
|---|---|---|
| N | 19 | 5 |
| Age range [years] | 19–35 | 18–55 |
| Visual Acuity (OU) | >20/25 | 20/10-20/115 |
| Spherical Equivalent [Median (Range)] | -0.13D (-2.00D to +1.00D) | +0.50D (-4.50D to +3.00D) |
| Residual astigmatism [Median (Range)] | 0.00D (0.00D to +1.00D) | -0.25 D (0.00D to +4.50D) |

| Binocularly impaired participants' details | | | |
|---|---|---|---|
| OD VA | OS VA | OU VA | Clinical detail |
| 20/13 | 20/33 | 20/15 | Left esotropia |
| 20/13 | 20/13 | 20/10 | Alternate Strabismus |
| 20/115 | 20/115 |  | Bilateral Amblyopia (high astigmatism) |
| 20/14 | 20/48 | 20/21 | Anisometropic Amblyopia |
| 20/12 | 20/100 | 20/10 | Anisometropic Amblyopia |

Top) Summary of demographics for Study 1 for both groups. Bottom) Details of all binocularly impaired participants. Residual spherical equivalent and astigmatism as determined via autorefraction.

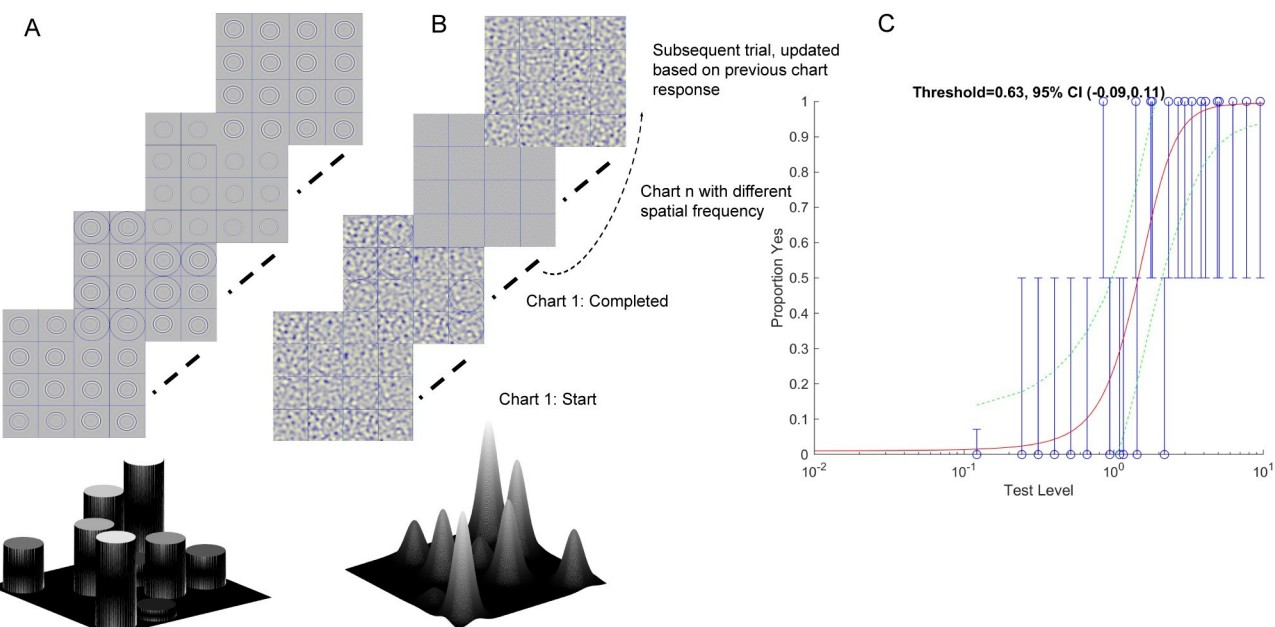

**Fig 1. FInD Stereo paradigm.** FInD Depth charts for A) Ring and B) Dip stimuli. Participants clicked the cells where they perceive targets in depth (i.e., in front (Ring) or behind (Dip) relative to the background). The range of disparities presented on each chart spanned easy (d' = 4.5) to difficult (d' = 0.1) and was adaptively calculated based on the participant's responses to previous charts. Depth profiles are shown for easy visualization for ring and dip stimuli at the bottom of the figure. C) The responses of the participant (blue circles, error bars indicate 95% binomial standard deviation) were used to calculate d' as a function of stereoscopic disparity and a decision function was used to estimate the probability of a Yes response (red curve). Green dashed lines indicate 95% confidence intervals at each stereoscopic disparity.

random deviate between 0.66N and N-1) contain a signal stimulus that vary from easy- to hard-to-detect intensity levels, the rest contain null stimuli, in random positions (Fig 1). The participant's task is to select cells that contain a signal. Three charts per condition, i.e. one initial and two adaptively changed charts, were deployed, each comprising 4*4 cells, each cell subtended 4°*4° with a 0.01° (1 pixel) black ($\approx$0 cd/m$^2$) border that also served as a fusion lock. Each cell contained either a signal (stereoscopic disparity $\neq$ 0) or null (stereoscopic disparity = 0) stimulus. The stimuli were either rings (2.5° radius, 1.5 arcmin line width; Fig 1A) or depth-defined Gaussian-shaped ($\sigma$ = 1°) dips within a noise carrier (Gaussian luminance distribution, element size 1 pixel; Fig 1B). The ring and dip stimuli investigate different aspects of stereopsis. The ring stimuli consist of sparse contour features, referred to as 'local' stereopsis, whereas the dip stimuli are defined by dense noise elements, referred to as 'global' stereopsis. These stimulus types have been used in different populations and with some evidence for separate processing mechanisms [23]. Using standard chart-based tests, stereoacuity estimates are similar for local and global stereopsis tests [24]. Ring and the noise carrier of dip stimuli were band-pass filtered with an isotropic raised log cosine filter:

$$H(f) = \begin{cases} \log_2(\omega) < \log_2(\omega_{peak}) - 1 = 0 \\ 0.5*(1 + cos(\pi*(\log_2(\omega) - \log_2(\omega_{peak})))) \\ \log_2(\omega) > \log_2(\omega_{peak}) + 1 = 0 \end{cases} \qquad [1]$$

where $\omega$ is spatial frequency and the peak spatial frequency was either 1, 2, 4, or 8 cycles/°. The Michelson contrast of the stimuli was scaled to 100%, with mean luminance 125 cd/m$^2$.

The magnitude of signal stereoscopic disparity on each chart was log-scaled from easy (d' = 4.5) to difficult (d' = 0.1), adaptively for each participant. On the first chart, the disparity range

was scaled to span 0.005˚ (0.3 arcmin) to 0.5˚ (30 arcmin) (which were also the upper and lower bound min and max disparity) in evenly spaced log steps to cover the broad typical stereoacuity range for binocularly healthy adults [25]. On subsequent charts, the disparity range was based on the results of the fit of Eq 2 to the data from all previous charts. For ring stimuli, stereoscopic disparity was created by horizontally displacing the ring in each eye by half the required disparity in opposite directions. For the Gaussian dip stimuli, stereoscopic disparity was created by generating spatial offsets in the noise carrier in opposite directions in each eye using spatial image warping as in [26] with a Gaussian profile (σ = 1˚). Ring stimuli had crossed disparity and the dip stimuli had uncrossed disparity.

Participants had unlimited time to click on cells that contained a target with depth and not on cells where the target contained no depth. Fig 1 shows the experiment procedure, once the participant had clicked a cell, a black circle appeared outside the target to indicate that the cell had been selected, and they could click an unlimited number of times to select or deselect (black circle disappeared) a response. Once they were satisfied with their selections, participants clicked on an icon to proceed to the next chart. The response in each cell was then classified as a Hit, Miss, False Alarm, or Correct Rejection, to calculate *d'* as a function of stereoscopic disparity, and the probability of a Yes response as a function of signal intensity was calculated as:

$$p(Yes) = 1 - \Phi\left( \Phi^{-1}(1 - F) - \frac{d'_{max} \times \left(\frac{S}{\theta}\right)^{\gamma}}{\sqrt{\left((d'_{max})^2 - 1\right) + \left(\frac{S}{\theta}\right)^{2\gamma}}} \right)$$

[2]

where *p(Yes)* is the probability of a Yes response, $\phi$ is the normal cumulative distribution function, *F* is the false alarm rate, *S* is stimulus intensity, $\theta$ is threshold, $d'_{max}$ is the saturating value of *d'* and was fixed at 5, and $\gamma$ is the slope. The fit to data from all completed charts was used to select the individualized range of stereoscopic disparities (from *d'* = 0.1 to 4.5) stimuli for subsequent charts.

## 2-AFC Stereo

To compare stereoacuity estimates of FInD with the gold standard psychophysical paradigm [27], the same participants completed 2 alternative forced choice (2-AFC) tasks. The stimuli were the same as in FInD, with signal and null stimuli presented either side-by-side for rings, or sequentially for dips. The ring stimuli had crossed disparity and the dip stimuli had uncrossed disparity. Fig 2 illustrates the experiment procedure and Fig 2A and 2B show the ring and dip stimuli. For the spatial 2-AFC ring procedure, the two stimuli were presented side-by-side for 1.25 sec and for the temporal 2-AFC procedure, the two dip stimuli were presented sequentially for 0.50 sec each, separated by a blank field for 0.50 sec. Participants indicated which of the two stimuli contained a depth target and they had unlimited time to respond. The different spatial frequency stimuli were randomly interleaved within separate runs for ring and dip stimuli. A 3-down-1-up algorithm [28] adjusted the stereoscopic disparity each trial with a total of 40 trials for each of the 4 spatial frequency conditions, resulting in 160 trials for ring and 160 trials for the dip stimuli per participant. The raw data were fitted with a cumulative normal function from which threshold was calculated as the 75% correct point.

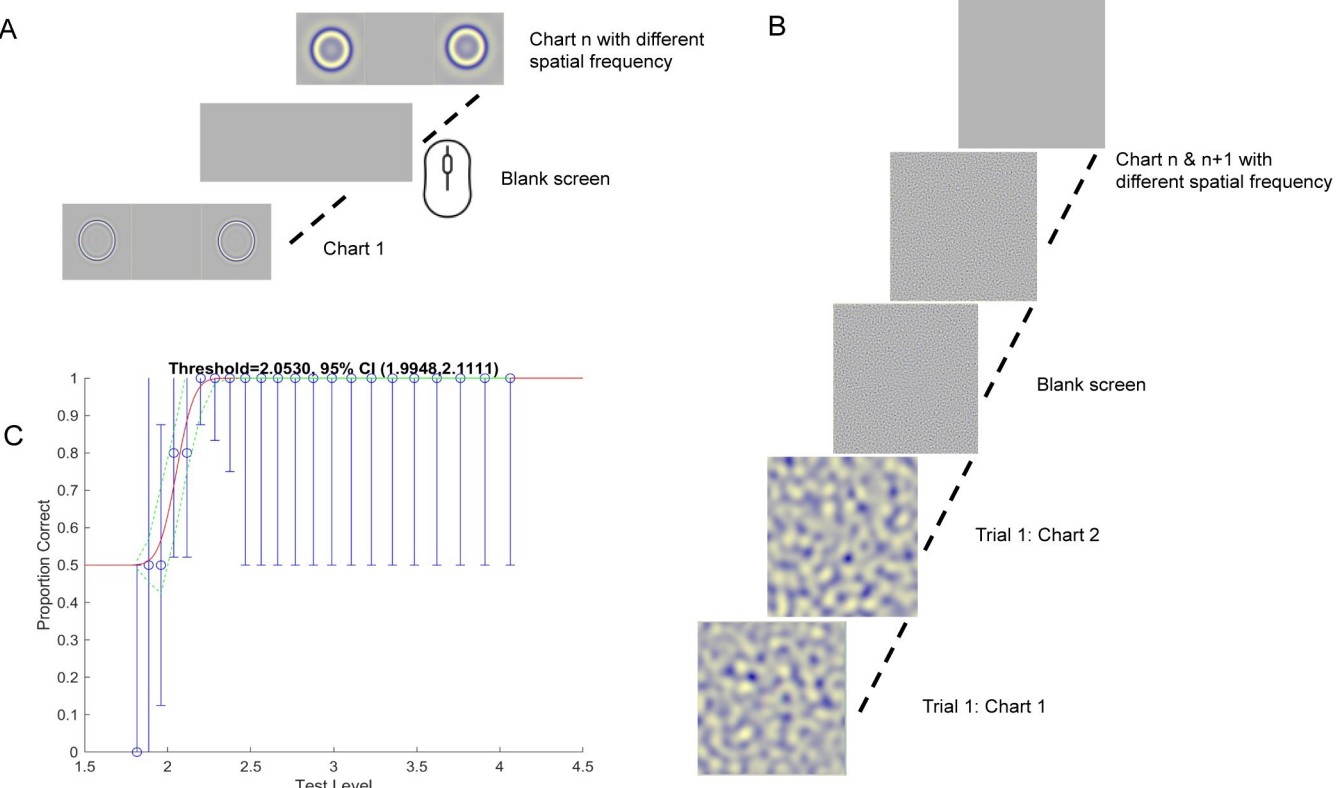

**Fig 2. AFC paradigm.** A) Spatial 2-AFC Ring task: target & null stimuli were presented side-by-side for 1.25 sec. Participants clicked on the left or right side of the screen to indicate whether the right or left stimulus was presented in depth. B) Temporal 2-AFC Dip task: target & null stimuli were presented sequentially for 0.50 sec each, separated by a blank screen for 0.50 sec. Participants clicked on the left or right mouse button to indicate whether the first or second stimulus contained depth. C) The proportion of correct trials as a function of stereoscopic disparity (blue circles, error bars indicate 95% binomial standard deviation) for each spatial frequency were fit with a cumulative gaussian function (red line) green dashed lines indicate 95% confidence limits at each disparity.

## Clinical tests -Titmus and Randot

The Titmus stereo test (Stereo Optical Company, Inc., USA) has 3 sections to measure stereoacuity at 40 cm: fly, Wirt circles (0.7° ∅), and animals. The fly section measures gross stereoacuity at 59 mins of arc, with pass or fail scoring. The circle and animal sections measure the stereo-threshold between 800–40 arcsec and 400–100 arcsec, respectively. The Randot stereo (Stereo Optical Company, Inc., USA) test also has 3 sections at 40 cm: circles, forms, and animals. The Wirt circles, forms, and animals sections measure the stereo-threshold between 400–20 arcsec, 500 arcsec and 400–100 arcsec respectively. The stereo threshold was taken as the highest stereoacuity the participant could observe on any section, which was found with the circles stimuli for most participants.

## Statistical analysis

Experiment duration and threshold estimates were analyzed with Matlab's *anovan* and *multcompare* functions for ANOVA and planned comparisons between tests. Duration data were skewed for FInD and Titmus data (Study 1) and AIM data (Study 2) and log-transformation was applied to convert durations to normally distributed data. Threshold estimates were log-transformed to convert the stereo-values to log-stereoacuity. The data for ring scotoma (FInD and 2AFC) and clinical tests (both Study 1 and 2) were still skewed, and further transformation

did not convert it to normally distributed data. Hence, Wilcoxon signed rank test and Kruskal-Wallis tests were performed for these threshold data. The *corrplot* function was used to compare linear correlations across tests, and stimulus conditions, using the Kendall's rank correlation coefficients. We used customized Bland-Altman plots to analyze the repeatability for all tests.

## Results—Study 1

### Test duration

We measured the test duration of the control and binocularly impaired participants with the FInD, 2-AFC, and the clinical methods. The test durations from two runs and 4 spatial frequencies were averaged for each participant (Fig 3). There was not a significant difference in test duration between the control participants and the binocularly impaired participants (p = 0.48). Overall, both the control and binocularly impaired participants took the least time with the clinical tests (mean 88.9 sec) followed by FInD tests (215.9 sec) and 2-AFC methods (316.9 sec) (F(1,5) = 25.34, p = 0.002). There was not a significant test duration difference between dip and ring stimuli (p>0.05) or between Titmus and Randot Stereotest (p>0.05).

**Stereo threshold.** Threshold stereoacuities at each test spatial frequency, are shown in Fig 4 using log-stereoacuity for control (blue data points) and binocularly impaired (magenta data points) participants measured with the FInD Ring, FInD Dip, 2-AFC Ring, and 2-AFC Dip tasks. Stereoacuities for Randot and Titmus tests are shown in Fig 4C and 4F. Each participant completed two assessments for each test and the mean of the thresholds was used in the analyses. Values greater than 3000 arcsec (3.48 log arcsec) were removed from the data analysis, resulting in the removal of 24 thresholds from 3 control participants (total: 608 thresholds) and 24 thresholds for 2 binocularly impaired participants (total: 152). Stereoacuities for all control participants were 60 arcsec or less with clinical tests and stereo-threshold measured

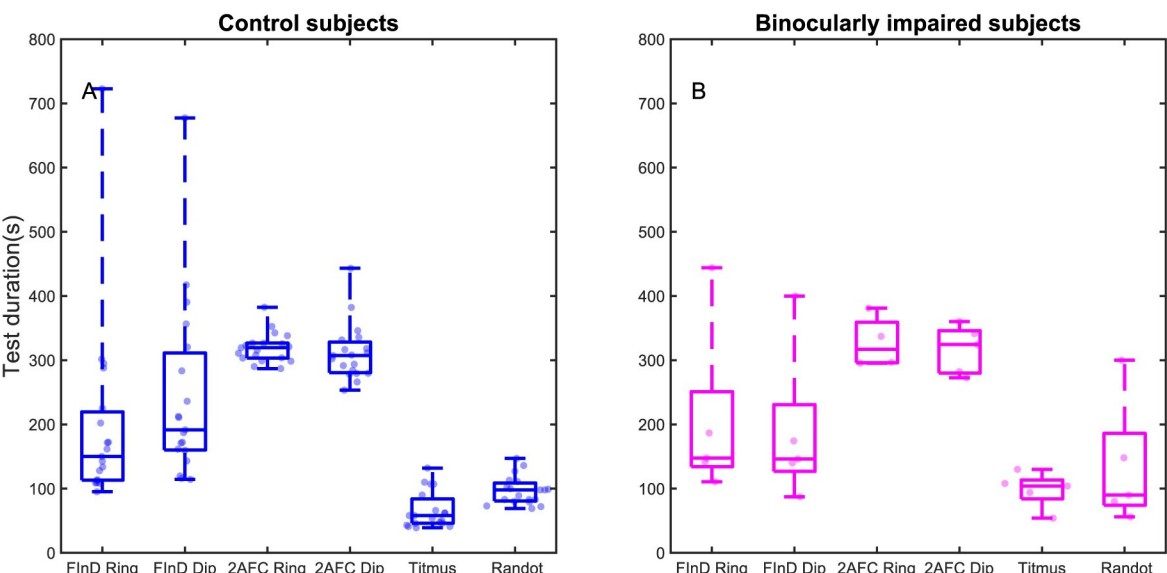

**Fig 3. Boxplots of test duration for stereoacuity assessment compared among tests.** Total Test Duration for FInD (3 charts for each of 4 spatial frequencies) for ring and dip stimuli, 2-AFC control experiments (40 trials for 4 interleaved spatial frequencies) for Ring and Dip stimuli, and the clinical tests Titmus and Randot. Test durations are shown in seconds for control (left panel, blue) and binocularly impaired (right panel, magenta) participants. Data points show the results for individual participants expressed by a horizontally jittered kernel density, boxes indicate the 25–75% interquartile range, whiskers represent 1st and 99th percentiles.

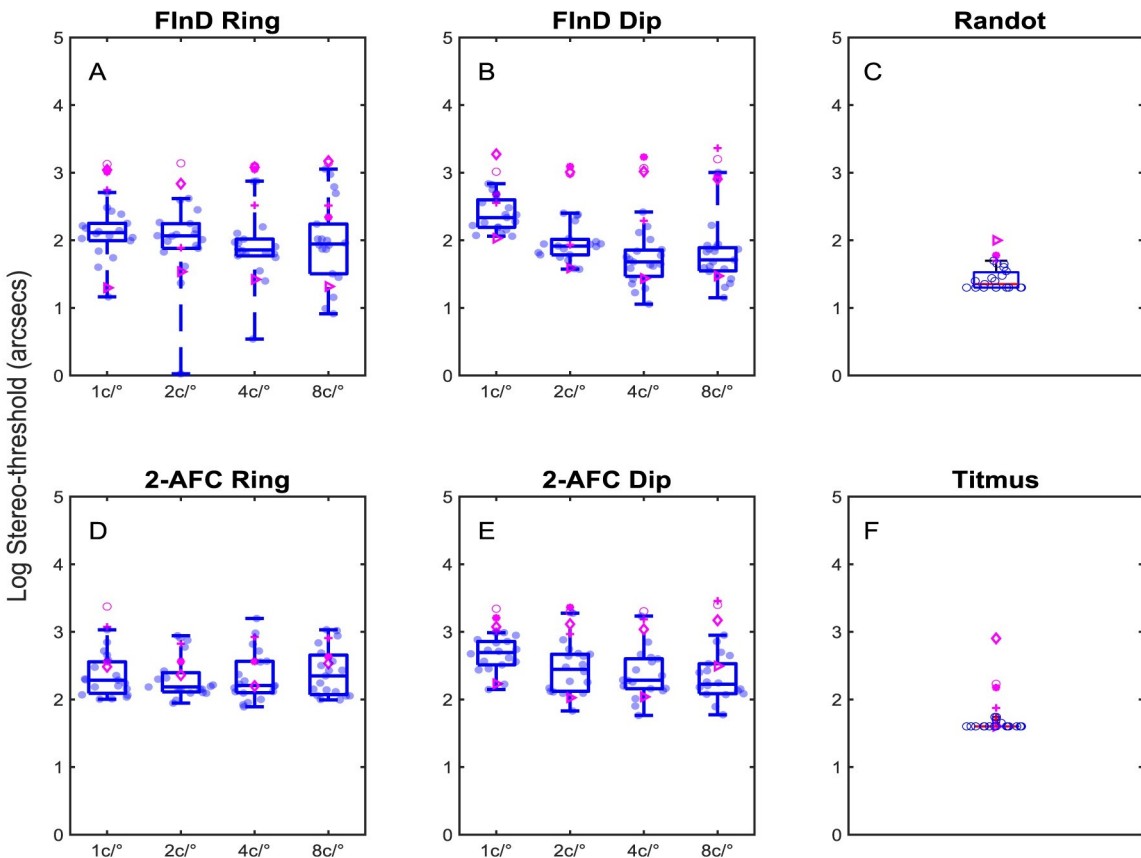

**Fig 4. Boxplots of log stereo-thresholds.** Stereoacuity thresholds in log arcsec are shown for A) FInD Ring B) FInD Dip D)2-AFC Ring E) 2-AFC Dip for each test peak spatial frequency 1, 2, 4, 8 c/° and for C) Randot F) Titmus clinical tests. Individual thresholds from control participants are shown as blue data points in the boxes, which indicate the 25–75% interquartile range, whiskers represent 1st and 99th percentiles. Data from participants with impaired binocularity are plotted in magenta and are not included in the boxplot calculations.

with FInD and 2-AFC tests were 1.3 and 1.6 times higher respectively than the clinical tests. The binocularly impaired participants had a wide range of stereoacuity in the clinical tests and this was observed with the other tests (compare control [blue] and binocularly impaired [magenta] data in Fig 4). The application of the data transformation failed to convert the skewed data to normally distributed data. For computer based tests, there was a significant difference in stereo-thresholds between group ($H(1,364) = 54.38$, $p<0.0001$; Kruskal-Wallis), and the overall test type ($H(3,362) = 56.76$, $p<0.0001$; Kruskal-Wallis), However, there was not a significant effect of spatial frequency($H(3, 362) = 1.98$, $p = 0.58$; Kruskal-Wallis).

Fig 5 shows the Kendall's rank coefficients of determination ($R^{2)}$) between Randot, Titmus, FInD and 2-AFC Stereo-tests of binocularly normal participants. 5 out of 15 correlations were significantly different from zero ($p<0.05$), and are identified in red. The Randot and Titmus thresholds correlated with each other but did not significantly correlate with any computerized task. FInD Dip and Ring correlated with each other, FInD Ring correlated also with both AFC generated thresholds. Both 2-AFC results also correlated with each other.

**Repeatability results.** Fig 6A and 6B show Bland-Altman analyses of repeatability between the same test from the control participants and the binocularly impaired participants. The results show that all the test paradigms have comparable test-retest repeatability with little bias.

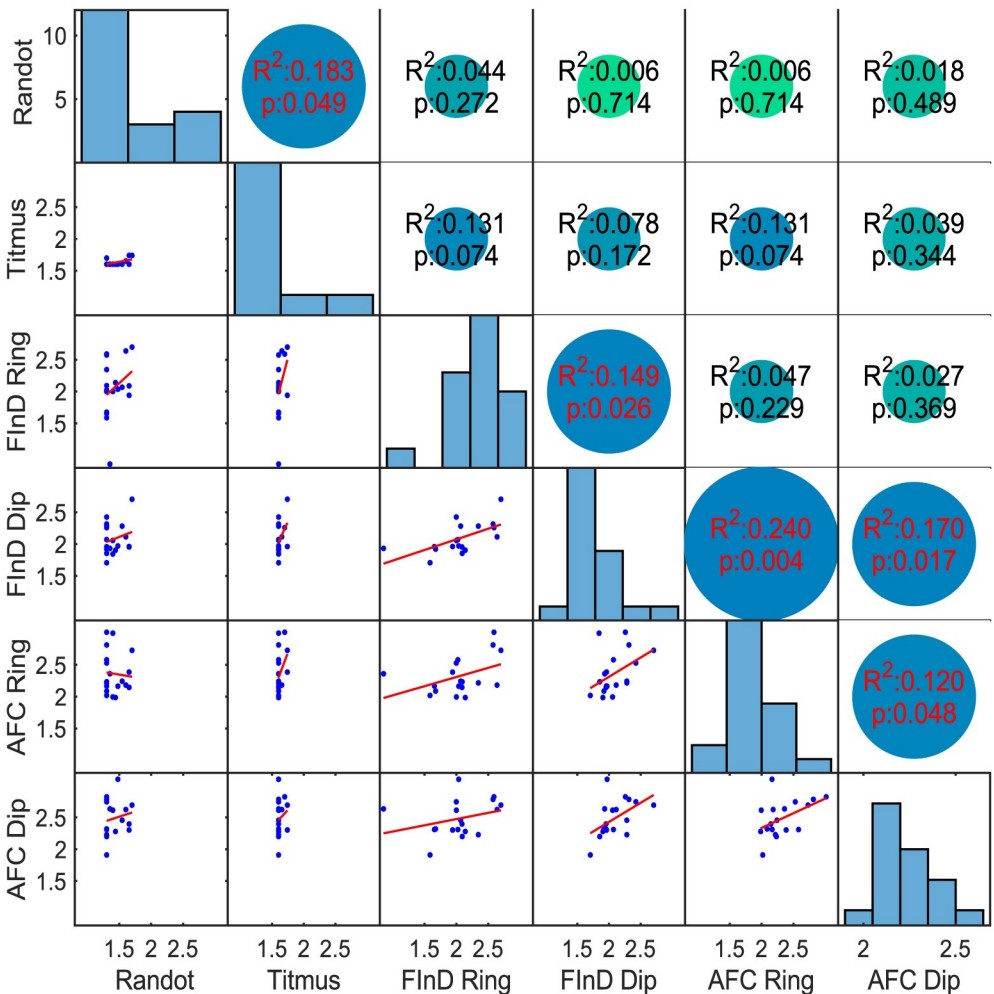

**Fig 5. Correlations between tests.** Kendall's rank correlations ($R^2$) expressed as linear function (red line) between log-stereoacuities generated by Randot, Titmus, FInD Ring and FInD Dip, and the 2-AFC Ring and Dip tests of binocular normally sighted participants on bottom left, and p-values indicated numerically in top right side of graph (red numbers refer to significantly different from null hypothesis). Histograms of data distribution for each test are shown in the diagonal. FInD and 2 AFC data are averaged across spatial frequency conditions.

## Discussion–Study 1

There was a low correlation among most of these tests, consistent with previous studies showing that most of the tests currently used in the clinics and research labs have low agreement. A study by Matsuo et al. (2014) found low correlation (0.43) between Titmus and TNO Stereoacuity [29]. Similarly, the correlation between the 3 rods test and Titmus/TNO/Distant Randot was less than 0.5 in Matsuo et al. (2014) study and 0.2 between 3 rod test and Distant Randot [29, 30]. In another study by McCaslin et al. (2020), the correlation between Randot and Asteroid was 0.54 whereas it was 0.66 between Randot and Propixx stimulus [31]. Vancleef et al. (2017) study also shows the low agreement between TNO and other stereo-tests [32].

A surprising aspect was the overall stereoacuity threshold estimates on the computer tests (2-AFC and FInD) were 1.3–1.6 times higher than the clinical tests. The stereo-threshold of the control participants were within 55 (1.74 log) arcsec with Randot and Titmus but with the 2-AFC and FinD, it was within 1260 (3.1 log) arcsec. There may be different reasons for this.

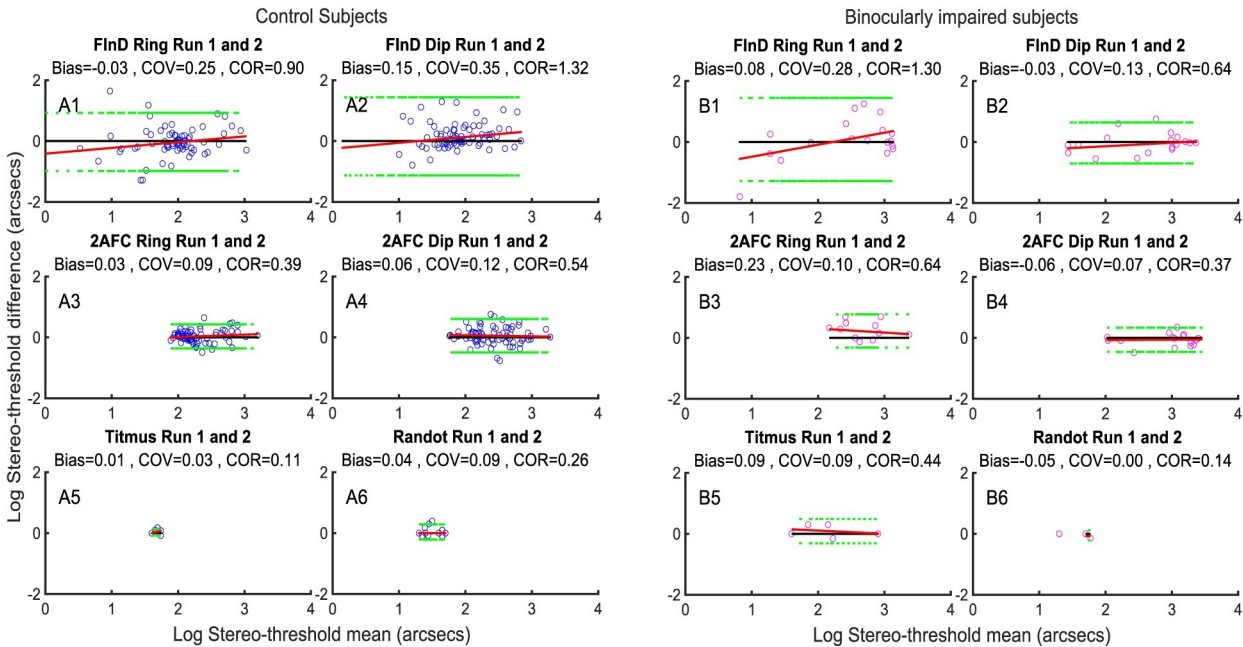

**Fig 6. Repeatability between tests.** Repeatability of FInD Ring, FInD Dip, 2-AFC Ring, 2-AFC Dip, Titmus and Randot tests from 2 runs for A) control participants B) binocularly impaired participants. Blue dots represent control participants (6A) and magenta dots represent binocularly impaired participants 6B). Each panel consists of 6 figures, corresponding to the FInD Ring, FInD Dip, 2-AFC Ring, 2-AFC Dip, Titmus and Randot tests. Each figure contains 1 data point for each participant for Randot and Titmus and 4 data points for each participant, one for each of the 4 spatial frequencies tested for FInD and 2-AFC.

One reason may be due to the pixel limitations of the screen. The display subtended 47.26° and each pixel subtended 0.74 arcmin/ 44.3 arcsec, which is close to the lowest stereoacuity (50 arcsec) measured in our computer tests, although we enabled subpixel rendering in Psychtoolbox. Secondly, studies by Hess and colleagues have suggested that many normally sighted people have poor stereopsis and that clinical standard tests are unable to detect these stereo anomalous populations, suggesting that monocular artefacts may lead to an overestimate of stereoacuity by clinical tests [25]. Thirdly, it may also be that the difference in the test protocol. Vancleef et al. (2017) have found stereoacuity thresholds are approximately 2 times higher for TNO than Randot [32], even greater than the difference we observe. They also hypothesize that the use of anaglyph red-green 3D glasses (used for our computer tests) may reduce binocular fusion and cause binocular rivalry more than the polarizing filters (used in the clinical tests). Other groups have reported that anaglyph glasses may increase stereoacuity error due to chromatic imbalance and rivalry [33]. The higher thresholds for 2AFC tests may be related to the fixed presentation time for 2AFC, while presentation time was unlimited for FInD, TNO than Randot. Fourthly, clinical tests report the smallest disparity that was correctly detected, whereas FInD and 2-AFC report thresholds at a specified criterion level above guessing rate ($d'$ = 1 or 75% correct, respectively), which may be higher than the lowest detectable disparity. Lastly, we, and some of the naïve participants, noticed that there was uncertainty about the reference depth of the display screen, even though there was a fusion box in all cases. This meant that both target and null stimuli in FInD cells and 2-AFC intervals sometimes appeared to be in depth relative to the display. This effect would increase the false alarm rate and decrease $d'$ in FInD paradigms and increase errors in 2-AFC tasks.

These various sources of error may have collectively contributed to the differences among tests. To address uncertainty concerning the presence of depth, we developed AIM Stereo as

**Table 2. Demographic and optometric summary of cohort for Study 2 experiments.**

|  | Binocularly normal participants |
| --- | --- |
| N | 21 |
| Age Range | 18–22 years |
| Visual Acuity (OU) | 20/20-20/30 |
| Residual Refractive error | -0.75 D to +1.13 D (Median: +0.19 D) (abs: +0.31 D) |
| Residual astigmatism | 0 to 1.25 D (Median: 0.25 D) |

an alternative forced choice paradigm that does not require a subjective estimate of a reference depth plane.

## Methods- Study 2

23 normally sighted adults participated in Study 2. Two participants were excluded as they were unable to complete the experiment (one because of headache and another due to technical error). Participants' details for Study 2 are provided in Table 2.

In Study 2, participants performed AIM Stereo, Randot, and Titmus tests in randomized order. The time taken for participants to complete the self-administered AIM tasks was recorded by the test computer. AIM Stereo was repeated twice on the same day.

### AIM Stereo

AIM is a self-administered paradigm in which stimuli are displayed in a series of charts comprising a grid of cells all of which contain a target [34]. For AIM Stereo, 3 charts, with 4*4 cells each of which contained 100 dots (0.14˚) within a 6˚ø circular area, surrounded by a white response ring with 0.1˚ line width, were deployed (see Fig 8A for example charts). Stereoscopic disparity was applied to dots within a 5˚ x 1.2˚ rectangular bar of random orientation within the noise background. This test was performed with a red-blue anaglyph display and glasses in crossed and uncrossed disparity sign. The stereoscopic disparity of each bar was selected to span a range from difficult (-2σ) to easy (+2σ) relative to a threshold-estimate that was selected by the experimenter (1˚ to 1') in chart one and thereafter based on a fit to data from previous charts (see Eq 3). The participants had unlimited time to report the orientation of the depth-defined bar via mouse click, guessing when they were unsure. Their reported orientation was displayed by two black ($\approx$0 cd/m$^2$) feedback marks, and they could adjust their report with further clicks. Once they had indicated the orientation of the bar in all cells, they could click on a 'Next' icon to proceed to the next chart. Orientation errors (i.e. the difference between indicated vs. actual bar-orientation) as a function of horizontal disparity were fit with a cumulative Gaussian function to derive stereo-thresholds:

$$\theta_{err} = \theta_{min} + (\theta_{max} - \theta_{min}) * \left( 0.5 - 0.5 * erf\left(\frac{\delta - \delta_\tau}{\sqrt{2}\gamma}\right) \right) \qquad [3]$$

Where $\theta_{err}$ is orientation error, $\theta_{min}$ is the minimum report error for a highly visible target, $\theta_{max}$ is the maximum mean error for guessing (here 45˚), $\delta$ is stimulus stereoscopic disparity, $\delta_\tau$ is threshold disparity and $\gamma$ is the slope. Thus, the threshold is derived from the midpoint between minimum angular error and 45˚ maximum mean error. Due to AIM's continuous report paradigm, the model enables a personalized performance profile and includes

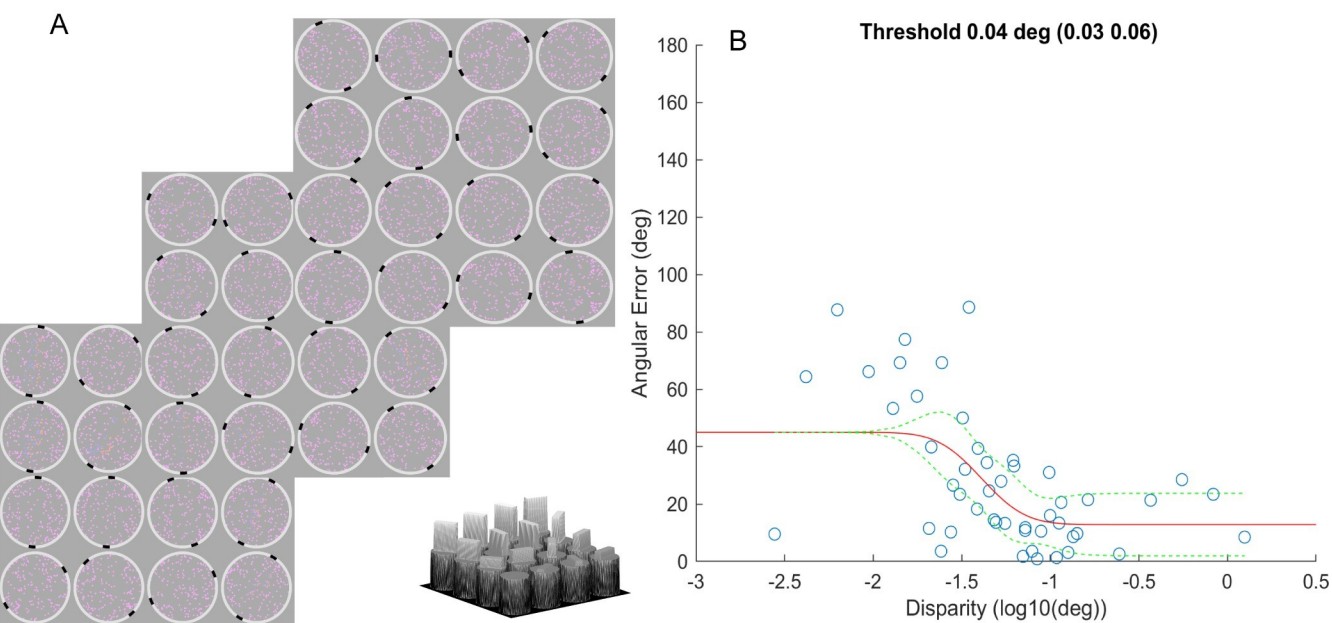

**Fig 7. AIM- Stereo paradigm.** A) Participants viewed three AIM charts, each containing a 4*4 grid of 6° ø cells with 100 dots, red in one eye and blue to the other, with a central disparity-defined 5°x1.25° rectangular bar of random orientation. Participants indicated the perceived orientation of the bar by clicking on the corresponding angle on the white ring surrounding each cell. Two black marks indicated the reported orientation and participants could adjust the reported orientation with unlimited further clicks. The range of disparities presented on subsequent charts was adaptively calculated based on their responses to previous charts. Visualization of the depth appearance of the stimuli is presented at the bottom of the figure. B) Angular error function (red line) using AIM paradigm. The y axis depicts the indicated orientation error for each disparity level (x axis). The responses of a representative participant's indications of each bar orientation error (blue circles) as a function of the stereoscopic disparity of the bar. The dashed green lines are 95% confidence intervals of the fit.

threshold, slope, and minimum angular error parameters. See Fig 7B for an example psychometric function.

## Results- Study 2

### Experiment duration

Fig 8 shows the test duration for the AIM task. For 3 charts of 4*4 grid of cells (48 orientation reports), the average duration for the first run was 241 sec and 117 sec for the second run, respectively. Log-transformation was applied before applying statistics to transform the skewed data to normally distributed data. This test time difference between runs was statistically significant (p<0.0001; paired t-test), suggesting a learning effect using AIM Stereo. When using 3 charts, i.e., one initial and two adaptive steps, AIM took significantly longer to complete than Titmus (66 sec) or Randot (97 sec), (F(2, 56) = 66.41, p<0.0001) from the Study 1 tests. However, post-hoc analysis of AIM Stereo using fewer charts (i.e. first chart only or first and second) shows that test time is significantly reduced (37 sec median first chart, 73 secs for first and second chart for the second run). The test time for first and second chart from second run of AIM is similar to Titmus but significantly shorter than Randot (F(2, 56) = 7.75, p<0.01).

### Stereo threshold

Fig 9 shows log stereo-threshold for Randot, Titmus and AIM. The median (inter-quartile range) stereo-threshold with the Randot, Titmus and AIM were 1.40(0.18), 1.60(0) and 1.82 (0.49) log arcsec respectively. The application of the data transformation failed to convert the skewed data to normally distributed data.

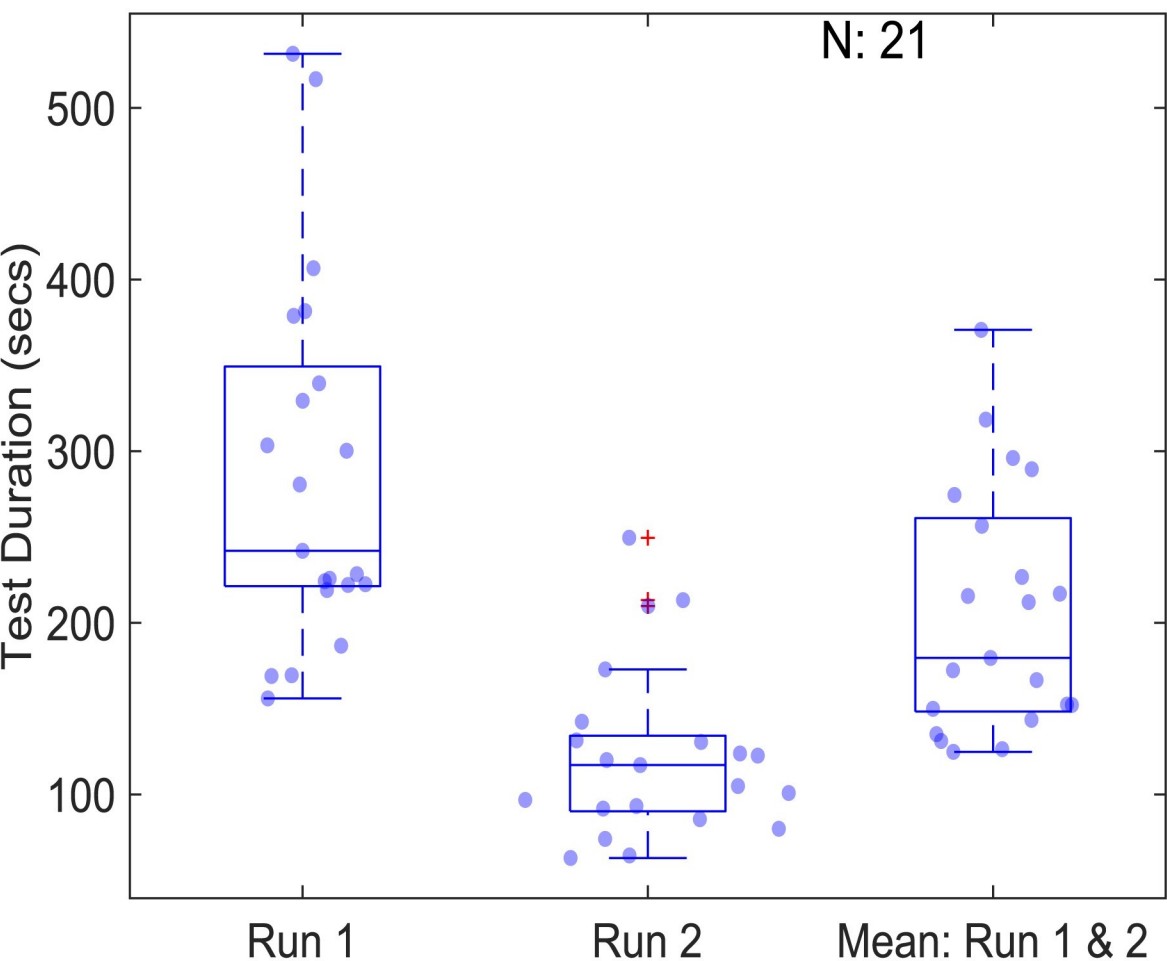

**Fig 8. Test duration using AIM Stereo for run 1 and run 2.** Data are plotted as in Fig 3A.

For AIM, the thresholds for the 2 runs were not statistically different (z(96) = -0.336, p = 0.73; Wilcoxon signed-rank test) and were averaged. The median of AIM stereo-thresholds were significantly higher (H(2,58) = 18.12, p = 0.0001; Kruskal-Wallis), than the stereo-thresholds from Randot or Titmus by a factor of 1.1–1.3 times.

Fig 10 shows Kendall's rank correlations (top-right triangle), histograms (diagonal) and linear functions (bottom-left triangle) between the Randot, Titmus, and AIM Stereo. All correlation between these tests were low and did not reach statistical significance ($R^2 \leq 0.038$, p <0.05). Histograms indicate a difference of distributions between stereo tests.

Fig 11 shows the repeatability of the AIM Stereo tests. There was a small bias (-0.01) and a tendency for an increase in estimated stereoacuity (decrease in stereo-threshold) on the second test, indicating a small learning effect. Overall, the stereo-thresholds tended to be stable over repetitions.

## General discussion

We introduce and evaluate two new stereoacuity tests, FInD and AIM Stereo, that were developed to address problems with current clinical tests. The results show that both FInD and AIM are comparable or faster in duration to current clinical tests (where all measure

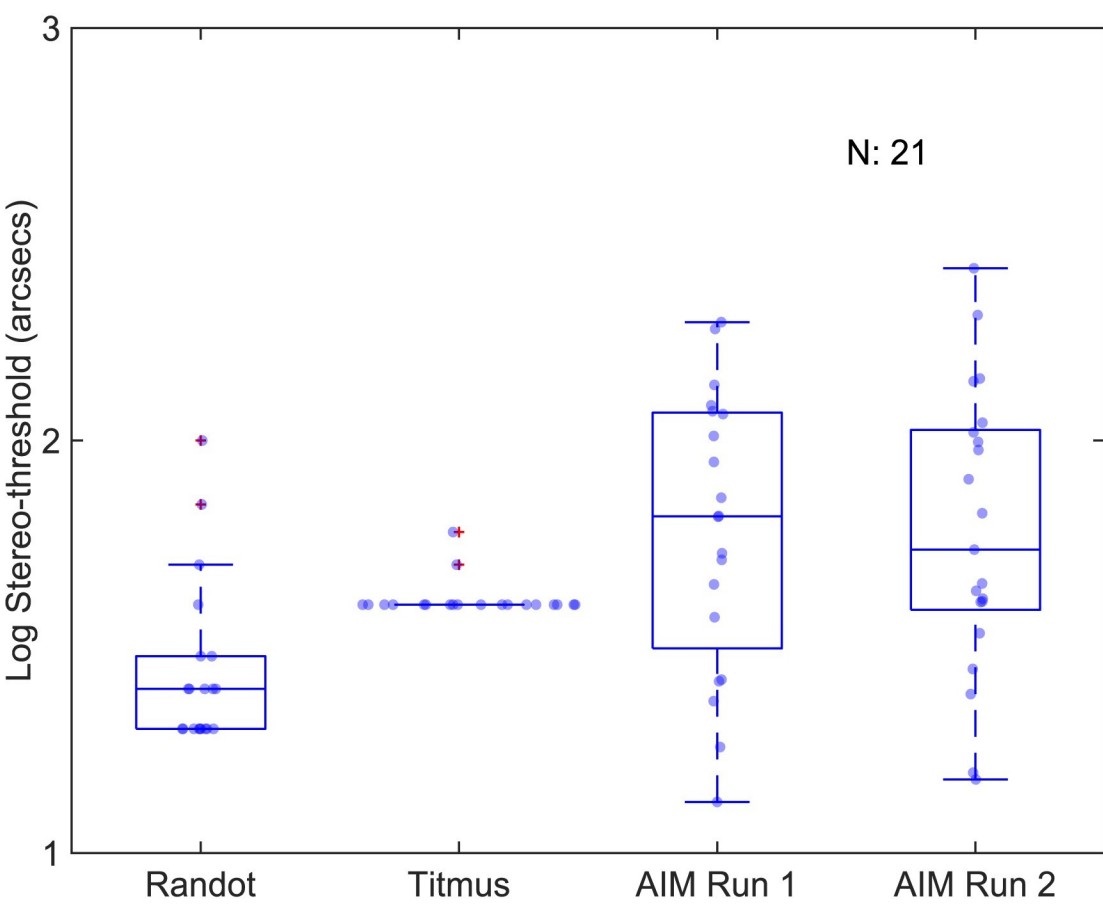

**Fig 9. Log stereoacuities of Randot, Titmus, and AIM Stereo.** Data are plotted as in Fig 4.

stereoacuity for a single spatial structure), are significantly faster than 2-AFC tests without loss of accuracy or precision and are sensitive to binocular visual impairment.

## Inter-Test agreement

There was good agreement between stereoscopic thresholds measured FInD and classic 2-AFC paradigms, suggesting that the faster speed of FInD did not come at the loss of accuracy. However, FInD and 2-AFC stereo thresholds were 1.3–1.6 times higher than those recorded by the clinical tests. We speculate that this difference could be attributed to underestimate of stereoacuity by FInD due to properties of the apparatus and task employed in the present study: the spatial resolution of our display (44 arcsec pixels) was close to the highest measured stereoacuity (50arcsec); the use of red/blue anaglyph stimuli could lead to rivalry that may impede binocular fusion [32]; differences in threshold criterion that were higher for FInD and 2-AFC than the smallest correct disparity for clinical tests; and uncertainty concerning the absolute depth of the reference plane may have led to false alarms. Additionally, the difference could be related to an overestimate of stereoacuity by clinical tests due to the presence of monocular artefacts.

While there was uncertainty concerning the target depth relative to the display in FInD stimuli, this ambiguity was eliminated with AIM in which a depth-defined bar is embedded in a disk of random dots. We speculate that this difference accounts for differences between

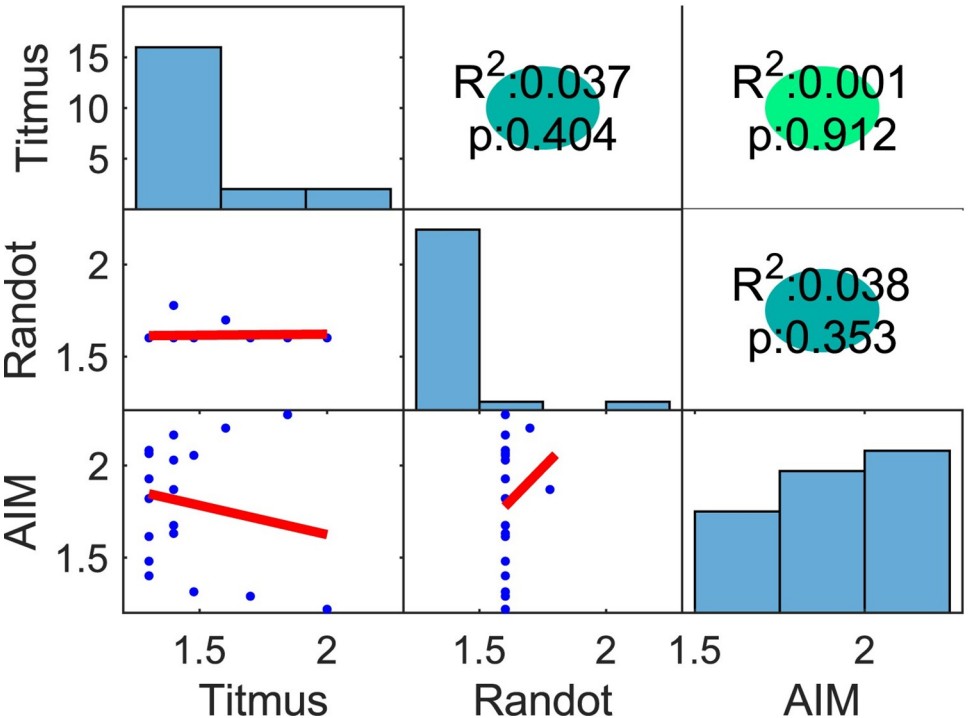

**Fig 10. Kendall's rank correlations and histograms between Randot, Titmus, and AIM Stereo.** Data are plotted as in Fig 5.

estimates of stereoacuity measured with clinical tests, which were 1.6 times lower than those measured with FInD, but only 1.1–1.3 times lower than those measured with AIM. This finding suggests that the test protocol, threshold criteria and stimulus parameters play a role in estimates of stereo-acuity, since anaglyph displays were employed by both AIM and FInD. The measurement of a psychometric function with FInD, AIM and 2-AFC paradigms also provides additional information next to the threshold parameter, including slope and, minimum error angle, which may provide useful information concerning sensitivity and bias [17].

## Repeatability

We investigated repeatability using Bland Altman plots and found no systematic learning effect as indicated as bias for FInD Stereo tests, 2-AFCs, or clinical tests for both controls and binocularly impaired individuals (Fig 6). The same was found for AIM Stereo (Fig 11). AIM Stereo showed less test-retest variability than FInD Stereo.

## Correlation analysis

In Study 1, the correlation ($R^2$) between Randot and Titmus was only 0.18 whereas in Study 2, it was 0.04. The correlation between clinical tests with other computer tests (FInD/2-AFC/AIM) ranged from 0.006 to 0.24. This variation frustrates efforts to compare results between studies that use different methods, and the same test must therefore be used to track recovery of stereoacuity.

## Stimulus structure

Surprisingly, there was no significant effect of spatial frequency on stereo thresholds measured with FInD or either spatial or temporal 2-AFC methods. This finding is not consistent with

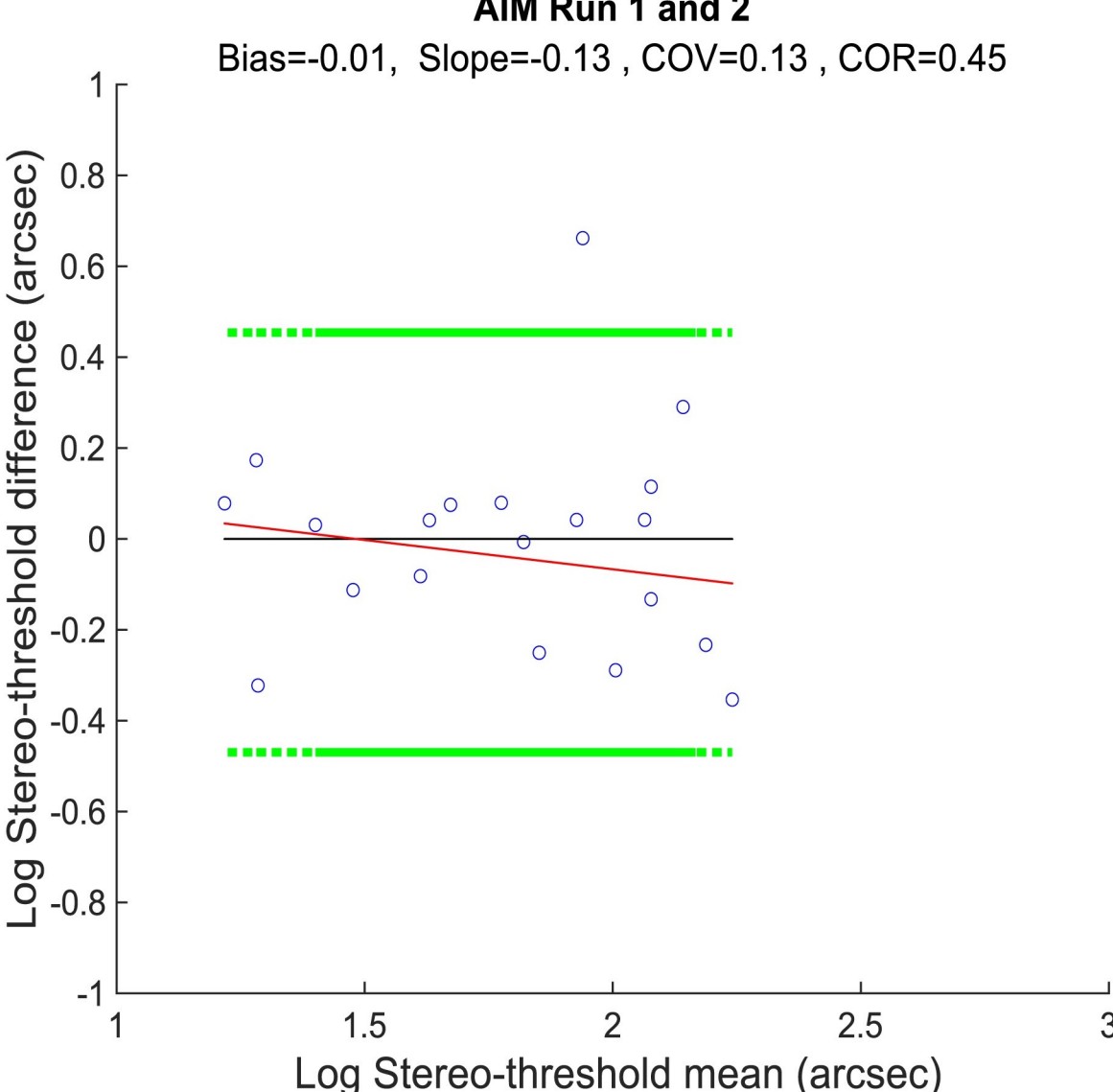

**Fig 11. Bland-Altman tests of AIM Stereo between the two runs.** Data are plotted as in Fig 6.

several previous studies [10, 11], however others have argued that the spatial scale of the disparity signal, which was constant in our stimuli, rather than the spatial frequency of the stimulus, determines the upper limit for stereoacuity [35]. Consequently, using fewer spatial frequency conditions will further shorten the testing time for the FInD Stereo tasks. Next to other methodological difference between the tests used in the current study, the stimulus property differences i.e., stimulus size, method of dichoptic presentation, spatial-frequency profiles, may have also contributed to the threshold differences between stereo-tests. One advantage of FInD, AIM, and 2-AFC tests is that they can be self-administered and thus can be potentially remotely used. This makes it easier to follow up any change of stereoacuity, that may occur due to therapy or disease progression, e.g., amblyopia therapy. This decreases the chair time in clinician's office while making it easier to keep track of the patient's change in stereoacuity threshold during progression or remediation of disease.

## Study limitations and future directions

A limitation of this study is the small number of binocularly impaired participants in the FInD experiment and none in AIM experiment. Another limitation is the lack of measurement of phoria. Phoria might impact the result by changing the depth plane relative to the background, particularly for the FInD paradigm where participants judged the presence of apparent depth in all cells. Some previous studies have shown that exophoric participants were likely to have better stereoacuity with crossed disparities and esophoric participants with uncrossed disparities when the phoria of more than 2 pd is considered [36–40]. Since we did not measure phoria in the present study, we cannot be sure that phoria did not differ across tests, or over time within tests, either of which could affect the results. We did not measure and compare crossed and uncrossed disparity within each FInD and AIM test, which will be a potential future direction. Although beyond the scope of the current study, increasing blank space between each cell may aid to establish a reference plane to judge depth, which may increase the repeatability of both FInD and AIM tests. The current study introduced AIM Stereo and compared its result with conventional clinical stereovision tests as proof-of-concept. We will compare AIM Stereo to other adaptive techniques, e.g., 2-AFC methods in future studies. AIM's approach offers two additional analysis features, namely a 3-parameter psychometric fit including threshold, slope, and min. angular report error and response error bias analysis [17], which may be suitable to detect distortions. A future investigation will examine whether these features provide additional psychometric biomarkers of stereovision impairment.

## Conclusions

In conclusion, this proof-of-concept study introduced the FInD and AIM Stereo methods and compared them with standard 2-AFC methods and clinical tests. The results reveal limitations across methods, including low agreement between tests but show promising results for FInD and AIM Stereo tests, which can be used as a self-administered metric to measure and monitor stereoacuity thresholds, accurately, precisely, quickly, remotely and over time. Different stimulus conditions did not significantly affect the thresholds of FInD Stereo. Also, FInD was able to detect atypical stereovision in participant with impaired binocular vision. AIM and FInD Stereo combine the stimulus control of classic psychophysics paradigms with the speed and ease-of-use of clinical tests while also adding additional analysis features as well as removing the need for a test administrator.

## Acknowledgments

Portions of this study have been presented at the Vision Science Society Conference 2022 and Vision Science Society conference 2023.

## Author Contributions

**Conceptualization:** Jan Skerswetat, Peter J. Bex.

**Data curation:** Sonisha Neupane, Jan Skerswetat, Peter J. Bex.

**Formal analysis:** Sonisha Neupane, Jan Skerswetat, Peter J. Bex.

**Funding acquisition:** Peter J. Bex.

**Investigation:** Sonisha Neupane.

**Methodology:** Sonisha Neupane, Jan Skerswetat, Peter J. Bex.

**Project administration:** Sonisha Neupane, Peter J. Bex.

**Resources:** Peter J. Bex.

**Software:** Sonisha Neupane, Jan Skerswetat, Peter J. Bex.

**Supervision:** Peter J. Bex.

**Validation:** Sonisha Neupane, Jan Skerswetat, Peter J. Bex.

**Visualization:** Sonisha Neupane, Jan Skerswetat, Peter J. Bex.

**Writing – original draft:** Sonisha Neupane.

**Writing – review & editing:** Sonisha Neupane, Jan Skerswetat, Peter J. Bex.

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
