## [Decision Letter · Decision Letter 0]

8 Jan 2024

PONE-D-23-27405Comparison of Foraging Interactive D-prime and Angular Indication Measurement Stereoacuity with different methods to assess stereopsisPLOS ONE

Dear Dr. Neupane,

Thank you for submitting your manuscript to PLOS ONE. After careful consideration, we feel that it has merit but does not fully meet PLOS ONE’s publication criteria as it currently stands. Therefore, we invite you to submit a revised version of the manuscript that addresses the points raised during the review process.

**ACADEMIC EDITOR: ** Both the reviewers have expressed concerns and provided some useful suggestions. Please address the comments thoroughly.

We look forward to receiving your revised manuscript.

Kind regards,

Amithavikram R Hathibelagal, Ph.D.

Academic Editor

PLOS ONE

Journal Requirements:

“This project was supported by National Institutes of Health (www.nih.gov)

(grant R01 EY029713 to PJB).”

4. We note that you have a patent relating to material pertinent to this article. Please provide an amended statement of Competing Interests to declare this patent (with details including name and number), along with any other relevant declarations relating to employment, consultancy, patents, products in development or modified products etc. Please confirm that this does not alter your adherence to all PLOS ONE policies on sharing data and materials, as detailed online in our guide for authors http://journals.plos.org/plosone/s/competing-interests by including the following statement: "This does not alter our adherence to  PLOS ONE policies on sharing data and materials.” If there are restrictions on sharing of data and/or materials, please state these. Please note that we cannot proceed with consideration of your article until this information has been declared.

6. We note you have included a table to which you do not refer in the text of your manuscript. Please ensure that you refer to Table1 & 2 in your text; if accepted, production will need this reference to link the reader to the Table.

Reviewers' comments:

Reviewer's Responses to Questions

**Comments to the Author**

1. Is the manuscript technically sound, and do the data support the conclusions?

Reviewer #1: Yes

Reviewer #2: Partly

2. Has the statistical analysis been performed appropriately and rigorously? 

Reviewer #1: No

Reviewer #2: Yes

3. Have the authors made all data underlying the findings in their manuscript fully available?

Reviewer #1: Yes

Reviewer #2: No

4. Is the manuscript presented in an intelligible fashion and written in standard English?

Reviewer #1: Yes

Reviewer #2: Yes

5. Review Comments to the Author

Reviewer #1: General Comments:

The study introduces two novel stereo measurement techniques, AIM and FlnD, and compares them with established tests such as 2AFC, Randot, and Titmus fly test. While the manuscript is technically sound, there needs a major restructuring with clarification and refinement for several points.

Clarity of Stimuli: The description of the FlnD noise stimulus needs further clarification. The term "noise" typically implies random dots without any signal. Consider renaming the target or providing a clear description of what subjects saw and responded to.

Viewing Distance: Ensure that details about the self-administrable tests, such as FlnD, AIM, and 2AFC, include information on how the viewing distance was maintained at 80 cm. Inconsistencies in viewing distance could contribute to data variation.

Aim of the study: Lines 137-144 outline the aims of the study but could be clarified. The lines starts out to stating there are 2 aims and explaining 5 aims. Consider explaining the features of FlnD and AIM in the introduction and focusing on the aims and objectives in this section.

Sections Organization: It is highly recommended that the FlnD and AIM results be combined as a single experiment for better coherence. The readers will be more interested in learning about the two paradigms rather than learning how the AIM came into being and to solve which problems of the FlnD paradigm. The author can discuss why AIM performed better than FlnD at the stereo thresholding in the discussion section.

Stats: Using stats to compare the results between the normal and binocular impaired individuals will lead to false results due to the huge uneven sample size

Specific Line-by-Line Comments:

Line 52: The classic sign for macular degeneration includes macular atrophy, drusens etc and the impact of these is loss of stereopsis. Stating stereopsis as a biomarker for degenerative diseases will be an error.

Line 115: Recheck the reference for the loss of stereoacuity due to refractive error and amblyopia as it refers to a paper related to retinal eccentricity.

Line 195-196: Provide consistency in units for the lower and upper bound min and max disparity, considering reporting in degrees.

Line 200-204: Clarify how the targets for the ring and noise stimuli were shifted towards opposite side and yet one produces crossed and another produces uncrossed disparities.

Line 275-276: Complete the sentence “Matlab’s anovan and multcompare functions for ANOVA and planned comparisons between tests”.

Line 290: Address the distribution of data in the statistical analysis section (if the data is normally distributed). Also, 2-way ANOVA was performed but looking at the fig 3, FlnD data, the mean and median are quite apart from each other.

Line 299: Clarify that whiskers usually represent 1st and 99th percentiles, not outliers.

Line 306: Explain the rationale for excluding values >3000 arcsec from analysis. Specially when they belong to the binocular impaired group, one would expect the stereo to be quite poor. If the data is skewed because of those subjects, a possibility would be to renumber stereo thresholds >3000 as 3000 and mention the reason for 3000 arcsec as the cutoff

Line 310-312: Discuss the potential implications of the observed variability

Line 314: p-value can never be exactly 0.

Line 345: “purple dot” – please keep colours consistent

Line 347: “Each figure contains 1 data point for each participant for Randot” shouldn’t the total number of dots be 20?

Line 406: Provide reasons for not including 2AFC in the comparison; consider combining the first and second experiments.

Line 457:Clarify the unit "5 X 1.25."

Line 457: It will be advisable to look at the learning in 2AFC and FlnD?

Line 489-499: Avoid repetition of introduction points and directly transition to conclusion statements.

Line 514: AIM was not tested on amblyopes in this manuscript.

Comments on the Figures:

Fig 3:

The different colours for the 6 tests are not required.

The horizontal jittering in 2AFC and clinical test seems to be huge. Please maintain uniform jittering.

Figure 4:

The normal can be given a single colour while the binocular impaired can be assigned a different colour and kept that uniform with figure 3. Also, number of subjects are not consistent.

Figure 5:

Consider combining with Figure 4 as panel E for easier comparison.

Figure 6:

Unequal number of subjects are presented here.

The graphs are repeated, for example randot Vs Titmus fly, or Titmus fly vs Randot will be same. Consider keeping only one to avoid confusion.

What additional inferences does the histogram generate?

Are these data points average of the 4 spatial frequency?

Consider including the Rsq value

Figure 7:

Align captions above the graphs for clarity.

Figure 11: Please remove repetitions of the graphs and mention the R sq and p values.

Reviewer #2: The study reports the inter-technique comparison and intra-technique repeatability of two new psychophysical measures of stereoacuity, relative to conventional clinical techniques and 2AFC psychophysical techniques. The two new tests were themselves quite repeatable but there was very little agreement between the various tests of stereoacuity. The time taken for the two new tests were shorter than conventional 2AFC techniques but significantly longer than the clinical tests of stereoacuity. These results are discussed in the context of a speed-accuracy trade-off in the assessment of stereoacuity.

The manuscript is easy to read and more or less free of grammatical errors. There are some confusing sentences dispersed all through the manuscript (indicated below in the specific comments) that need to be fixed. The figures are all of very low resolution and the details were not visible upon magnification. This needs to be fixed as well. In addition to these issues, the following issues need to be addressed in the manuscript.

1. While I do not have any concern with the quality of the science pursued in this manuscript, I worry about the utility of these techniques in the future assessment of stereoacuity in the clinic/research settings. This concern is more for the clinical application, and it primarily stems from the time it takes to complete the test, vis-à-vis, the present techniques of Titmus or RanDot stereoacuity. I appreciate that the two new techniques are far more scientifically rigorous, and they overcome several limitations fraught with the present testing paradigms. However, is this a motivation enough for an average clinician to switch over to this technique? These techniques take ~3-times the time to complete than the present techniques and in a busy clinic, this translates into significant chair time for the patient. I fear that the accuracy benefit obtained using this technique may be over-ridden by the time taken to complete the task.

There may be a solution to this though. As indicated in the discussion section of the manuscript (lines 506 – 509), testing only one spatial frequency stimulus or a single disparity sign may reduce the time to complete the task. Given that the target spatial frequency did not have an impact on the stereoacuity anyways, can this not be implemented in the present version of the software and evidence provided that this indeed reduces the time to do the task? I understand that this involves additional data collection and more work for the authors, but this may be a significant step towards the holy-grail of having a test that is both accurate and can be administered quickly.

2. The stimuli used in the two new stereoacuity techniques is not clear to me. There are bandpass filtered ring and noise stimuli that the participants had to appreciate depth in. What was the pattern of depth though? I assume that in the ring stimuli, the “ring” appeared in depth depending on the stimulus disparity used but what was seen in the “noise” stimuli? This is not clear from the explanation. Perhaps a figure showing the two stimuli may be useful to include here for helping the readers understand the stimulus.

3. Other specific issues…

a. Abstract conclusion: Why only FlnD detected stereo deficits. It is also detected by AIM right? Please rephrase.

b. Abstract conclusion: The last statement is a rather superficial one. It should be replaced with a statement that describes the advantage/challenges of switching over to the new techniques proposed in this study, vis-a-vis, the existing ones. The new techniques take ~3times longer to complete than the routine clinical tests. What would be the motivation for the clinician to switch over to the new technique should come out clearly here.

c. Line 78: Add to this list that these tests are meant to be run at a constant viewing distance and are meant for a single IPD - the former limits their ability to check stereopsis at different viewing distances while the latter creates issues with test accuracy.

d. In general, the introduction is way too long (4.5 pages). This can be significantly shortened, without losing information.

e. Line 123: Expand HMD’s.

f. Lines 137 - 144: This section is a bit confusing. The start makes it appear as if there are only two experiments, but the remainder of the manuscript appears like there are five parts to the study. So, add a statement before first to indicate that what ensues is a note on the organization of the manuscript.

g. Line 170: I am sure this was taken care of, but a brief mention that there was no leak of information between the red/blue filters to reassure the readers. Also, how was the issue of blur arising from chromatic aberrations of the eye handled in these stimuli?

h. Line 195: Should this be log arc sec?

i. Lines 286 – 287: Please include a measure of variance of the data in all measures. Also, please fix the p=0.000 to p<0.001.

j. Line 292: Use of Box plots suggests that data were not normally distributed. Was that the case? If so, the results need to be presented as median and IQR.

k. Line 517: Be explicit that the new tests are faster, relative to the 2AFC paradigms.

l. Like the introduction section, the discussion section could also be tightened. In several areas of this section, the results are repeated, and this redundancy can be removed.

6. PLOS authors have the option to publish the peer review history of their article (what does this mean?). If published, this will include your full peer review and any attached files.

Reviewer #1: No

Reviewer #2: No

---

## [Author Response · Author response to Decision Letter 0]

27 Feb 2024

Northeastern University

105-107 Forsyth St, 

Boston, MA, 02115

neupanesonisha@gmail.com

The Academic Editor

Investigative Ophthalmology & Vision Science

Feb 22, 2024

Dear Editor, 

We thank you and the reviewers for your time and comments on our manuscript. In this letter, we provide a detailed response to the points raised by the reviewers. 

We have highlighted the reviewers’ comments in bold, our responses are in blue plain text. We have also included the original and modified text where applicable and we have used italics for the lines from the manuscript.

We have revised the funding statement and conflict of interest as follows:

Funding Statement

This project was supported by National Institutes of Health (www.nih.gov)

(grant R01 EY029713 to PJB). There was no additional external funding received for this study. The funders had no role in study design, data collection and analysis, decision to publish, or preparation of the manuscript.

Competing Interests:

I have read the journal's policy and the authors of this manuscript have the following competing interests:

FInD and AIM technologies are disclosed as provisional patented (AIM) and pending patent (FInD) and held by Northeastern University, Boston, USA.

FInD title: Method for visual function assessment; Application PCT/US2021/049250

AIM title: Self-administered adaptive vision screening test using angular indication.

Application PCT/US2023/012959.

JS and PJB are founders and shareholders of PerZeption Inc, which has an exclusive license agreement for FInD and AIM with Northeastern University. SN declares that she has no conflict of interest. 

This does not alter our adherence to PLOS ONE policies on sharing data and materials.

Yours sincerely, 

Sonisha Neupane 

Journal Requirements:

We have edited the manuscript according to PLOS ONE’s style requirements.

We have deposited the data and the MATLAB code in the Zenodo repository. The DOI for our data and code is 10.5281/zenodo.10688863.

“This project was supported by National Institutes of Health (www.nih.gov)

(grant R01 EY029713 to PJB).”

This project was supported by National Institutes of Health (www.nih.gov)

(grant R01 EY029713 to PJB). There was no additional external funding received for this study. The funders had no role in study design, data collection and analysis, decision to publish, or preparation of the manuscript.

4. We note that you have a patent relating to material pertinent to this article. Please provide an amended statement of Competing Interests to declare this patent (with details including name and number), along with any other relevant declarations relating to employment, consultancy, patents, products in development or modified products etc. Please confirm that this does not alter your adherence to all PLOS ONE policies on sharing data and materials, as detailed online in our guide for authors http://journals.plos.org/plosone/s/competing-interests by including the following statement: "This does not alter our adherence to PLOS ONE policies on sharing data and materials.” If there are restrictions on sharing of data and/or materials, please state these. Please note that we cannot proceed with consideration of your article until this information has been declared.

I have read the journal's policy and the authors of this manuscript have the following competing interests:

FInD and AIM technologies are disclosed as provisional patented (AIM) and pending

patent (FInD) and held by Northeastern University, Boston, USA.

FInD title: Method for visual function assessment; Application PCT/US2021/049250

AIM title: Self-administered adaptive vision screening test using angular indication;

Application PCT/US2023/012959.

JS and PJB are founders and shareholders of PerZeption Inc, which has an exclusive

license agreement for FInD and AIM with Northeastern University. SN declares that she has no conflict of interest. This does not alter our adherence to PLOS ONE policies on sharing data and materials.

We have deposited the data and the MATLAB code in the Zenodo repository. The DOI for our data and code is the 10.5281/zenodo.10688863.

6. We note you have included a table to which you do not refer in the text of your manuscript. Please ensure that you refer to Table1 & 2 in your text; if accepted, production will need this reference to link the reader to the Table.

We have referred the Table 1 and 2 in the text.

XXXXXXXXXXXXXXXXXXXXXXXXXXXXXXXXXXXXXXXXXXXXXXXXXXXXXXXXXXXXXXXXXXXXXXXXXXXXXXX

Reviewers' comments:

Reviewer's Responses to Questions

Comments to the Author

1. Is the manuscript technically sound, and do the data support the conclusions?

Reviewer #1: Yes

Reviewer #2: Partly

2. Has the statistical analysis been performed appropriately and rigorously?

Reviewer #1: No

Reviewer #2: Yes

3. Have the authors made all data underlying the findings in their manuscript fully available?

Reviewer #1: Yes

Reviewer #2: No

4. Is the manuscript presented in an intelligible fashion and written in standard English?

Reviewer #1: Yes

Reviewer #2: Yes

XXXXXXXXXXXXXXXXXXXXXXXXXXXXXXXXXXXXXXXXXXXXXXXXXXXXXXXXXXXXXXXXXXXXXXXXXXXXXXX 

5. Review Comments to the Author

Reviewer #1: General Comments:

The study introduces two novel stereo measurement techniques, AIM and FlnD, and compares them with established tests such as 2AFC, Randot, and Titmus fly test. While the manuscript is technically sound, there needs a major restructuring with clarification and refinement for several points.

We thank the reviewer for taking the time to evaluate our paper and addressed each of the comments. To address the reviewers request for clarification and to improve the legibility of the revision, we changed the structure of all sections. We hope that the reviewer finds, that due to these changes, the overall legibility and quality of the manuscript has improved.

Clarity of Stimuli: The description of the FlnD noise stimulus needs further clarification. The term "noise" typically implies random dots without any signal. Consider renaming the target or providing a clear description of what subjects saw and responded to.

We thank the reviewer for the suggestion and have added the stimulus depth figures and edited the text to make it clear. We have also changed the name from ‘noise’ to ‘dip’ e.g. FInD Dip to avoid confusion. 

Viewing Distance: Ensure that details about the self-administrable tests, such as FlnD, AIM, and 2AFC, include information on how the viewing distance was maintained at 80 cm. Inconsistencies in viewing distance could contribute to data variation.

We have included the following line in the Method Section.

Line 124: “A chinrest was used to maintain the viewing distance.”

Aim of the study: Lines 137-144 outline the aims of the study but could be clarified. The lines starts out to stating there are 2 aims and explaining 5 aims. Consider explaining the features of FlnD and AIM in the introduction and focusing on the aims and objectives in this section.

We have edited the text to make it clear that two experiments were conducted to address five research aims.

Line 93-100:” The two experiments reported here were performed as proof-of-concept studies with five aims: First, we introduce FInD Stereo and its features. Second, we compare estimates of stereoacuity, test duration, and test-retest reliability of FInD with standard 2-AFC methods, and clinically used tests (Randot and Titmus) in stereo-typical and atypical participants. Third, we compare different spatial properties of stereo-inducing stimuli and examine their effect on stereoacuity. Fourth, we introduce AIM Stereo and its features. Fifth, we compare AIM Stereo against the above-mentioned clinical tests using the same outcome measures.”

Sections Organization: It is highly recommended that the FlnD and AIM results be combined as a single experiment for better coherence. The readers will be more interested in learning about the two paradigms rather than learning how the AIM came into being and to solve which problems of the FlnD paradigm. The author can discuss why AIM performed better than FlnD at the stereo thresholding in the discussion section.

The FInD and AIM were done at different times with different set of participants. To reflect this, we have kept them as two experiments, but we have edited the text to make the flow better.

Stats: Using stats to compare the results between the normal and binocular impaired individuals will lead to false results due to the huge uneven sample size

We appreciate the comment and acknowledge this limitation of our study accordingly.

Line 523-524: A limitation of this study is the small number of binocularly impaired participants in the FInD experiment and none in AIM experiment.

Specific Line-by-Line Comments:

Line 52: The classic sign for macular degeneration includes macular atrophy, drusens etc and the impact of these is loss of stereopsis. Stating stereopsis as a biomarker for degenerative diseases will be an error.

We have removed the section stating stereopsis as a biomarker for degenerative disease. 

Line 115: Recheck the reference for the loss of stereoacuity due to refractive error and amblyopia as it refers to a paper related to retinal eccentricity.

We have updated the reference.

Line 195-196: Provide consistency in units for the lower and upper bound min and max disparity, considering reporting in degrees.

We have changed the text to report them in degrees. 

Line 153-156: On the first chart, the disparity range was scaled to span 0.005° (0.3 arcmin) to 0.5° (30 arcmin) (which were also the upper and lower bound min and max disparity) in evenly spaced log steps to cover the broad typical stereoacuity range for binocularly healthy adults.

Line 200-204: Clarify how the targets for the ring and noise stimuli were shifted towards opposite side and yet one produces crossed and another produces uncrossed disparities.

It depends on which side of the disparities the eyes are seeing. If the right side targets are visible to right eye and left side to left eye, it produces the uncrossed disparities whereas if it is the reverse, it provides the crossed disparities. 

Line 275-276: Complete the sentence “Matlab’s anovan and multcompare functions for ANOVA and planned comparisons between tests”.

We have changed the sentence accordingly.

Line 235-236: Threshold estimates were analyzed with Matlab’s anovan and multcompare functions for ANOVA and planned comparisons were used to study the variance between tests.

Line 290: Address the distribution of data in the statistical analysis section (if the data is normally distributed). Also, 2-way ANOVA was performed but looking at the fig 3, FlnD data, the mean and median are quite apart from each other.

Some of the outcomes are not normally distributed. For data from Fig 3, we did log transformation to convert them to normally distributed data before performing ANOVA. The log threshold data (Figure 4) was not able to convert to normally distribution after further transformation and we have stated in the manuscript stating so. 

Line 245-247: Log-transformation was applied before using ANOVA test to transform the skewed data to normally distributed data.

Line 275-276: The application of the data transformation failed to convert the skewed data to normally distributed data.

Line 299: Clarify that whiskers usually represent 1st and 99th percentiles, not outliers.

We have changed the text to clarify that.

Line 259-261: Data points show the results for individual observers expressed by a horizontally jittered kernel density, means are depicted in dark squares, boxes indicate the 25-75% interquartile range, whiskers represent 1st and 99th percentiles.

Line 306: Explain the rationale for excluding values >3000 arcsec from analysis. Specially when they belong to the binocular impaired group, one would expect the stereo to be quite poor. If the data is skewed because of those subjects, a possibility would be to renumber stereo thresholds >3000 as 3000 and mention the reason for 3000 arcsec as the cutoff.

The Titmus fly is equal to 3552’’ threshold but has variable disparity in

---

## [Decision Letter · Decision Letter 1]

16 Apr 2024

PONE-D-23-27405R1Comparison of Foraging Interactive D-prime and Angular Indication Measurement Stereo with different methods to assess stereopsisPLOS ONE

Dear Dr. Neupane,

Thank you for submitting your manuscript to PLOS ONE. After careful consideration, we feel that it has merit but does not fully meet PLOS ONE’s publication criteria as it currently stands. Therefore, we invite you to submit a revised version of the manuscript that addresses the points raised during the review process.

**ACADEMIC EDITOR: ** Please address the minor comments raised by one of the reviewers.

We look forward to receiving your revised manuscript.

Kind regards,

Amithavikram R Hathibelagal, Ph.D.

Academic Editor

PLOS ONE

Journal Requirements:

Reviewers' comments:

Reviewer's Responses to Questions

**Comments to the Author**

1. If the authors have adequately addressed your comments raised in a previous round of review and you feel that this manuscript is now acceptable for publication, you may indicate that here to bypass the “Comments to the Author” section, enter your conflict of interest statement in the “Confidential to Editor” section, and submit your "Accept" recommendation.

Reviewer #1: All comments have been addressed

Reviewer #2: All comments have been addressed

2. Is the manuscript technically sound, and do the data support the conclusions?

Reviewer #1: Yes

Reviewer #2: Yes

3. Has the statistical analysis been performed appropriately and rigorously? 

Reviewer #1: Yes

Reviewer #2: Yes

4. Have the authors made all data underlying the findings in their manuscript fully available?

Reviewer #1: Yes

Reviewer #2: Yes

5. Is the manuscript presented in an intelligible fashion and written in standard English?

Reviewer #1: Yes

Reviewer #2: Yes

6. Review Comments to the Author

Reviewer #1: Really appreciate the effort taken to incorporate the changes into the manuscript.

The explanation of the aims is still not proper yet. What do the authors mean by two experiments? Was it the development of the FlnD and AIM? The author might want to avoid saying it's two experiments.

Figure 1 is impressive as it explains clearly, what the subjects would have been able to see. However, that makes me wonder why would the authors wanted to create these two stimuli. Authors might want to add a statement to indicate the pros of the ring and dip stimulus.

Lines 244 to 247 are a part of statistical analysis. Also in that section, it will be better to explain which variables were not normally distributed and that the authors have log-transformed the variable.

Usually, the F number is associated with two degrees of freedom not a single like F(5) and many more in the further text.

Fig 5 Are the correlation values Pearson’s correlations? Also, correlation is denoted with the symbol r. In the last review, when I suggested reporting the r2 values, I meant only for those instances where the correlation value was higher and the regression line could be fit and could have reported the r2 values then. In the same context, Line 39 in the abstract should say correlation instead of agreement.

Reviewer #2: The authors have addressed all the comments of the reviewers. The manuscript is much better organized now. I have nothing further to add to this review.

7. PLOS authors have the option to publish the peer review history of their article (what does this mean?). If published, this will include your full peer review and any attached files.

Reviewer #1: No

Reviewer #2: **Yes: **Shrikant R Bharadwaj

---

## [Author Response · Author response to Decision Letter 1]

20 May 2024

Comments to the Author

Reviewer #1: Really appreciate the effort taken to incorporate the changes into the manuscript.

We thank the reviewer for the appreciation.

The explanation of the aims is still not proper yet. What do the authors mean by two experiments? Was it the development of the FlnD and AIM? The author might want to avoid saying it's two experiments.

Thank you for the suggestion, we originally described the development of the FlnD and AIM as 2 groups of experiments. However, to avoid confusion, we have changed the presentation into Study 1 and 2, one for each novel method, and each with experiments that were completed to investigate stereoacuity with each method. The 2 Studies are differentiated by the main novel method (FInD or AIM) and by different subject pools.

Line 91-99: The two studies reported here were performed as proof-of-concept studies for 2 novel methods to measure stereoacuity: In Study One, we introduce FInD Stereo and its features. We compare estimates of stereoacuity, test duration, inter-test reliability, and repeatability of FInD with standard 2-AFC methods, and clinically used tests (Randot and Titmus) in stereo-typical and atypical participants. We also use the FInD method to compare different spatial properties of stereo-inducing stimuli and examine their effect on stereoacuity. In Study Two, we introduce AIM Stereo and its features, then we compare AIM Stereo against the above-mentioned clinical tests using the same outcome measures.

Figure 1 is impressive as it explains clearly, what the subjects would have been able to see. However, that makes me wonder why would the authors wanted to create these two stimuli. Authors might want to add a statement to indicate the pros of the ring and dip stimulus.

We have added a statement in methods to indicate that the ring and dip stimulus investigate different aspects of stereopsis.

Line 144-148: The ring and dip stimuli investigate different aspects of stereopsis. The ring stimuli consist of sparse contour features, generally referred to as ‘local’ stereopsis, whereas the dip stimuli are defined by dense noise elements, generally referred to as ‘global’ stereopsis. These stimulus types have been used in different populations and with some evidence for separate processing mechanisms.

Lines 244 to 247 are a part of statistical analysis. Also in that section, it will be better to explain which variables were not normally distributed and that the authors have log-transformed the variable.

We have edited the statistical analysis section to include the following: 

Line 241-248: Duration data were skewed for FInD and Titmus data (Study 1) and AIM data (Study 2) and log-transformation was applied to convert durations to normally distributed data. Threshold estimates were log-transformed to convert the stereo-values to log-stereoacuity. The data for ring scotoma (FInD and 2AFC) and clinical tests (both Study 1 and 2) were still skewed, and further transformation did not convert it to normally distributed data. Hence, Wilcoxon signed rank test and Kruskal-Wallis tests were performed for these threshold data. 

We have also changed the following lines in results section to reflect this:

Line 289-293: For computer based tests, there was a significant difference in stereo-thresholds between group (H(1,364)=54.38, p<0.0001; Kruskal-Wallis), and the overall test type (H(3,362)=56.76, p<0.0001; Kruskal-Wallis), However, there was not a significant effect of spatial frequency(H(3, 362)=1.98, p=0.58; Kruskal-Wallis).

Line 457-460 : For AIM, the thresholds for the 2 runs were not statistically different (z(96)=-0.336, p=0.73; Wilcoxon signed-rank test) and were averaged. The median of AIM stereo-thresholds were significantly higher (H(2,58)=18.12, p=0.0001; Kruskal-Wallis), than the stereo-thresholds from Randot or Titmus by a factor of 1.1-1.3 times. 

Usually, the F number is associated with two degrees of freedom not a single like F(5) and many more in the further text.

We have included the F number with two degrees of freedom wherever applicable.

Fig 5 Are the correlation values Pearson’s correlations? Also, correlation is denoted with the symbol r. In the last review, when I suggested reporting the r2 values, I meant only for those instances where the correlation value was higher and the regression line could be fit and could have reported the r2 values then. In the same context, Line 39 in the abstract should say correlation instead of agreement.

We used Kendall’s rank correlation and have noted that in the Statistical analysis section. We have added that in text and figure legend of Fig 5 and 10. 

Following the reviewer’s comments in the first round, we report r2 values in the revised manuscript. We use r2 uniformly throughout the manuscript without assigning a criterion for the use of r or r2.

Reviewer #2: The authors have addressed all the comments of the reviewers. The manuscript is much better organized now. I have nothing further to add to this review.

We thank the reviewer for the appreciation.

---

## [Editor Report · Decision Letter 2]

23 May 2024

Comparison of Foraging Interactive D-prime and Angular Indication Measurement Stereo with different methods to assess stereopsis

PONE-D-23-27405R2

Dear Dr. Neupane,

We’re pleased to inform you that your manuscript has been judged scientifically suitable for publication and will be formally accepted for publication once it meets all outstanding technical requirements.

Kind regards,

Amithavikram R Hathibelagal, Ph.D.

Academic Editor

PLOS ONE

---

## [Editor Report · Acceptance letter]

29 May 2024

PONE-D-23-27405R2 

PLOS ONE

Dear Dr. Neupane, 

I'm pleased to inform you that your manuscript has been deemed suitable for publication in PLOS ONE. Congratulations! Your manuscript is now being handed over to our production team.

Kind regards, 

on behalf of

Dr. Amithavikram R Hathibelagal 

Academic Editor

PLOS ONE